# Value of the Lymphocyte Transformation Test for the Diagnosis of Drug-Induced Hypersensitivity Reactions in Hospitalized Patients with Severe COVID-19

**DOI:** 10.3390/ijms241411543

**Published:** 2023-07-17

**Authors:** Carlos Fernández-Lozano, Emilio Solano Solares, Isabel Elías-Sáenz, Isabel Pérez-Allegue, Monserrat Fernández-Guarino, Diego Fernández-Nieto, Laura Díaz Montalvo, David González-de-Olano, Ana de Andrés, Javier Martínez-Botas, Belén de la Hoz Caballer

**Affiliations:** 1Biochemistry-Research Department, Hospital Universitario Ramón y Cajal, Instituto Ramón y Cajal de Investigación Sanitaria, Carretera de Colmenar Km 9, 28034 Madrid, Spain; carlitos4mb@hotmail.com; 2Allergy Service, Hospital Universitario Ramón y Cajal, Instituto Ramón y Cajal de Investigación Sanitaria, Carretera de Colmenar Km 9, 28034 Madrid, Spain; emilio.solano.solares@gmail.com (E.S.S.); isaes014@gmail.com (I.E.-S.); ipallegue@gmail.com (I.P.-A.); lauragdm97@gmail.com (L.D.M.); dgolano@yahoo.es (D.G.-d.-O.); 3Dermatology Service, Hospital Universitario Ramón y Cajal, Instituto Ramón y Cajal de Investigación Sanitaria, Carretera de Colmenar Km 9, 28034 Madrid, Spain; montsefdez@msn.com (M.F.-G.); fnietodiego@gmail.com (D.F.-N.); 4Immunology Service, Hospital Universitario Ramón y Cajal, Instituto Ramón y Cajal de Investigación Sanitaria, Carretera de Colmenar Km 9, 28034 Madrid, Spain; aandresm@salud.madrid.org

**Keywords:** delayed drug hypersensitivity, skin reaction, SARS-CoV-2, LTT, interleukins

## Abstract

In the first wave of COVID-19, up to 20% of patients had skin lesions with variable characteristics. There is no clear evidence of the involvement of the SARS-CoV-2 virus in all cases; some of these lesions may be secondary to drug hypersensitivity. To analyze the possible cause of the skin lesions, we performed a complete allergology study on 11 patients. One year after recovery from COVID-19, we performed a lymphocyte transformation test (LTT) and Th1/Th2 cytokine secretion assays for PBMCs. We included five nonallergic patients treated with the same drugs without lesions. Except for one patient who had an immediate reaction to azithromycin, all patients had a positive LTT result for at least one of the drugs tested (azithromycin, clavulanic acid, hydroxychloroquine, lopinavir, and ritonavir). None of the nonallergic patients had a positive LTT result. We found mixed Th1/Th2 cytokine secretion (IL-4, IL-5, IL-13, and IFN-γ) in patients with skin lesions corresponding to mixed drug hypersensitivity type IVa and IVb. In all cases, we identified a candidate drug as the culprit for skin lesions during SARS-CoV-2 infection, although only three patients had a positive drug challenge. Therefore, it would be reasonable to recommend avoiding the drug in question in all cases.

## 1. Introduction

SARS-CoV-2 is a respiratory virus that can affect multiple organs, causing a wide range of symptoms in some patients [1]. Cutaneous involvement, in which many types of skin lesions are identified [2], was described in the first published papers on SARS-CoV-2. COVID-19 skin reactions were found to be generally higher in Western Europe than in Asia, with 6.6% reported in Europe compared with 0.2% in Asia [3]. Initially, an Italian group described six types of skin lesions: maculopapular rashes, urticarial rashes, vesicular rashes, erythema multiforme, cutaneous vasculitis, and chilblain-like lesions [4]. These lesions were considered secondary to the infection, but hypersensitivity to the treatments received could not be ruled out with absolute certainty [5]. The lesions described in patients with SARS-CoV-2 infection were very heterogeneous and had a similar pattern to those observed in delayed drug hypersensitivity reactions (e.g., maculopapular exanthema (MPE) and fixed drug eruption (FDE)), drug-induced liver injury (DILI), and severe cutaneous adverse reactions (SCARs) (e.g., Stevens–Johnson syndrome (SJS), toxic epidermal necrolysis (TEN), drug reactions with eosinophilia and systemic symptoms (DRESSs), and acute generalized exanthematous pustulosis (AGEP)) [6,7,8,9]. The most common drugs prescribed for COVID-19 treatment were hydroxychloroquine (18.5%), azithromycin (11.1%), lopinavir (7.4%), ritonavir (7.4%), and paracetamol (9.2%) [3].

Late skin reactions to drugs belong to a mechanism of type IV hypersensitivity mediated by T cells. Advances in our knowledge of the cells and cytokines involved in these types of reactions have allowed them to be classified into four types (IVa-IVd) [10]. Type IVa corresponds to T-helper type 1 (Th1) cytokine-driven responses associated with high IFN-γ/TNF-α secretion. Type IVb corresponds to T-helper type 2 (Th2) cytokine-driven reactions with the increased secretion of IL-4, IL-5, and IL-13. Type IVc corresponds to the cytotoxic reactions mediated by cytotoxic CD8 T cells and seems to be the primary mechanism of bullous skin reactions, such as SJS and TEN. Type IVd represents the T-cell-induced sterile neutrophilic inflammatory response, e.g., AGEP [11].

Diagnostic tests for delayed drug hypersensitivity are scarce. In vivo tests like epicutaneous patches are the most readily available. These patches must be prepared with the suspected drug involved in the reactions (based on a very detailed allergy clinical history) with an appropriate concentration and vehicle in order to yield accurate results. A positive result confirms the involvement of the drug, but the predictive value of a negative test is unknown. Therefore, a drug challenge is still considered the gold standard for diagnosing a drug allergy. In most delayed reactions, this option is not possible due to the patient’s risk of reaction and the lack of standardization of the challenged drugs, as a complete single dose of a drug may rule out an immediate IgE-mediated reaction but not a delayed reaction that may occur after consecutive doses in a longer treatment.

In recent years, the lymphocyte transformation test (LTT) has been used to diagnose delayed drug-induced hypersensitivity reactions by detecting the proliferation of drug-specific memory T cells [12]. In addition, previous studies have shown that the measurements of cytokine secretion in PBMCs may be useful in diagnosing drug hypersensitivity [11,13,14].

Our group conducted a prospective, observational, and descriptive study to determine whether drug hypersensitivity was the real cause of skin lesions. The results have been reported previously [15]. The aim of the present study was to confirm the mechanism of hypersensitivity and the drugs involved in the skin lesions observed in patients with SARS-CoV-2 infection by means of an immunological study.

## 2. Results

### 2.1. Design and Setting

Patients were selected from our previous study, which was a prospective, observational, and descriptive study for which its main objective was to determine whether drug hypersensitivity could have been a cause of skin lesions in patients admitted to our hospital with SARS-CoV-2 infection between March and May 2020 [15]. Of the 72 patients included in this study, 37 were classified as having a possible drug-caused lesion according to the Spanish Pharmacovigilance System (ASPS) [16]. Of these, only 16 agreed to continue in the study. In all cases, a complete allergological study was performed via skin tests, epi-patches, and oral challenges against the drugs used during the period of infection and skin lesions [15]. In the present study, 11 of these 16 patients agreed to finish the “in vitro” study (Figure 1). We also included five nonallergic patients (NAPs) who were exposed to these drugs but did not develop lesions. The main treatments used were dolquine (hydroxychloroquine (HCQ)); azithromycin (AZT); kaletra (lopinavir/ritonavir (LOP/RIT)); and/or beta-lactam antibiotics, such as amoxicillin/clavulanic acid (AMOX/CLA) or ceftriaxone (Table 1).

The patients included in this study presented three types of cutaneous lesions: maculopapular exanthema (MPE), urticarial exanthema (UEX), and vesicular exanthema (VEX). Accordingly, patients were classified as generalized exanthema (*n* = 10) or cutaneous vasculitis (*n* = 1) (Table 1). In all cases, the time from the start of treatment to the onset of lesions was between 1 and 15 days, with a mean of 7.5 days. Consistently with the general characteristics of the most severe COVID-19 patients, 75% of the patients were male and had a median age of 62 years (IQR 71-58.75).

Epicutaneous patch tests were performed 4–6 months after hospital discharge, with a negative result for the eleven patients. In the case of beta-lactam antibiotics, the skin prick test with late-reaction lecture ware was also performed, and all results were negative. Afterward, a drug provocation test (DPT) was performed with the implicated drugs on alternative days. DPTs were performed in 9 out of the 11 patients. Two patients had no exposure to a DPT; one had no exposure because of the severity of his initial lesions due to cutaneous vasculitis, and the other refused the DPT. The DPT results were positive in three patients: two for AZT (late maculopapular exanthema and vesicular exanthema) and one for AMOX/CLA (macular exanthema). It is important to mention that patient P2 presented an immediate reaction to 12.5 mg of AZT. In all three cases, the cutaneous lesions were consistent with the initial ones during COVID-19 treatment (Table 1).

### 2.2. Lymphocyte Transformation Test (LTT)

In order to identify the possible culprit drugs causing skin lesions in these patients, we performed an LTT one year after recovery from COVID-19. An LTT was also performed in five nonallergic patients. The study was conducted one year after recovery from COVID-19. In all cases, the LTT was performed using three doses of AZT, AMOX, CLA, HCQ, LOP, and RIT (Table 2). Ceftriaxone was excluded from the LTT study due to the limited sample size and the results of the allergology study. The test was considered positive with a stimulation index ≥ 3. Except for patient 2, who had an immediate reaction to AZT, all patients had a positive LTT result for at least one of the drugs tested. Three patients tested positive for AZT (P5, P6, and P10). Of these, patient 5 also had a positive DPT. One patient tested positive for CLA with a positive DPT (P3). Three patients tested positive for HCQ (P1, P5, and P9). Four patients tested positive for LOP (P4, P6, P7, and P8). Seven patients tested positive for RIT (P4, P6, P7, P8, P9, P10, and P11). None of the patients responded to AMOX. Two patients responded to only one drug, and eight responded to two drugs. None of the nonallergic patients had a positive LTT result for any of the drugs tested. All results are shown in Table 2.

### 2.3. Cytokine Secretion

Next, we analyzed the cytokine secretion of PBMCs in response to all relevant drugs four days after drug stimulation. In patients with cutaneous lesions, cytokine release was measured in all conditions with a positive LTT result and at least one negative drug. In the case of patient 2, who had an immediate reaction to AZT, cytokine secretion was analyzed at all concentrations of this drug (0.1, 1, and 10 μg/μL) and at 5 μg/μL of RIT. In the group of nonallergic patients without cutaneous lesions, cytokine secretion was analyzed at a representative concentration for each drug except for AZT, which did not give any positive LTT results. In the group of patients with cutaneous lesions, the LTT-positive drugs strongly induced the secretion of IL-4, IL-5, and IL-13 (Figure 2 and Appendix A) in most patients. The levels of these cytokines in the nonallergic patient’s group were consistently low, and no increase was observed with any of the selected drugs. Although there was an apparent increase in IFN-γ with the LTT-positive drugs, an increase in IFN-γ was also observed in three nonallergic patients, which occurred with all tested drugs. It is interesting to note that patient 2, who had an immediate response to AZM and a negative LTT result, had very low levels of these cytokines. The response of the other cytokines studied, IL-1β, Il-6, TNF-α, and IL-10, was inconsistent, as they increased with some treatments but not others. These cytokines were also increased by some non-proliferation-stimulating drugs and in control patients.

## 3. Discussion

During the first wave of the COVID-19 pandemic, up to 20% of patients had skin lesions of different characteristics [4,17,18]. The skin lesions associated with COVID-19 were classified into six categories: maculopapular exanthems, urticarial exanthems, vesicular exanthems, erythema multiforme, cutaneous vasculitis, and chilblain-like lesions [4]. Due to the heterogeneity of the treatments, it has not been possible to clearly establish whether or not some of the skin lesions that were presented in patients during the first wave of COVID-19 could be secondary to drug hypersensitivity. In the present study, we analyzed the possible cause of skin lesions during SARS-CoV-2 infection in 11 patients. Using lymphocyte proliferation and cytokine secretion assays, we identified a drug candidate as the culprit despite only three patients having positive drug provocation test results.

The fact that skin manifestations were greatly reduced in subsequent waves when COVID-19 treatments were changed supports the view that the skin manifestations observed in the first wave were mainly due to hypersensitivity reactions to the drugs used at that time [19] or to the combination of both the drugs used and the viral strain in the subsequent waves of COVID-19. Such interactions between drugs and viral infections have been widely reported for the viruses of the Herpesviridae family and less commonly for other viruses such as influenza, chikungunya, or HIV [7].

An LTT is recommended for the diagnosis of drug hypersensitivity reactions (DHRs) in which the distal effector phase is mediated by T cells [12]. After the PBMC culture, the activation of the lipocytes begins within minutes due to a specific drug antigen presented by major histocompatibility complex (MHC) class I or II antigen-presenting cells (APCs). Following T-cell receptor (TCR) activation, Ca^2+^ increases, and a signaling cascade activates early antigen recognition genes. Over the next few hours, the expression of genes encoding several cytokines (IL-2, 3, 4, 5, and 6; IFN-γ; and TGF-β) and early activation markers increases. One to two days after T-cell activation, IL-2 induces the proliferation of activated T-cells; consequently, DNA synthesis starts. Approximately three to five days after activation, T cells enter the functional differentiation phase and produce different cytokine patterns: Th1, Th2, or Th3. Th1 is mainly associated with the production of IL-2, IFN-γ, and TNF-α; Th2 is associated with the production of IL-4, IL-5, and IL-13; Th3 is associated with the production of IL-17A and IL-17F. The type of specific T cell produced depends on the sensitization phase, and these cytokine patterns determine the effector functions of T lymphocytes [11]. Several studies have shown that cytokine secretion in the supernatant of drug-stimulated PBMCs may also be useful in the diagnosis of drug hypersensitivity [13,14]. The production of Th1 cytokines, mainly IL-2, IFN-γ, and TNF-α, in PBMCs has been associated with DHR in several studies [11]. High IFN-γ production by drug-stimulated PBMCs has been observed during the acute allergic phase in SCARs such as SJS, TEN, DRESS, or AGEP [20]. IL-5 increases in patients with drug-induced MPE and DRESS [21,22], and it has been proposed as a useful in vitro method for detecting drug sensitization. Furthermore, the combination of IL-5 measures and the LTT may better indicate drug sensitization than the LTT alone [23]. Other studies have shown a mixed Th1/Th2 cytokine pattern with the production of IL-5, IL-4, and IL-13 in addition to IFN-γ. Indeed, high levels of IL-5 and IFN-γ secretion by CD4 cells are associated with maculopapular exanthema [10] and have been proposed as promising in vitro indicators of drug hypersensitivity [11,24]. Lochmatter et al. [13] extensively studied the secretion of 17 cytokines and chemokines in PBMCs from patients with well-documented drug allergies. They found that the measurement of IL-5 combined with IFN-γ, IL-13, or IL-2 is the more sensitive marker for detecting T-cell sensitization to drugs.

Consistently with this, we found mixed Th1/Th2 cytokine secretion (IL-4, IL-5, IL-13, and IFN-γ) in patients with skin lesions. Therefore, this allows us to classify these patients as having mixed type IVa and IVb drug hypersensitivity reactions, corresponding to T-helper type 1 (Th1) cytokine-driven responses that are associated with high levels of IFN-γ secretion and Th2 cytokine-driven responses that are associated with high levels of IL-4, IL-5, and IL-13 secretion.

It is important to note that we found elevated levels of IFN-γ in some patients without skin lesions, a phenomenon that has been described previously [25,26].

Among the eleven patients studied, an immediate clinical response was observed in only one case, which was confirmed by a positive oral challenge and a negative LTT result for AZT. In the remaining cases, a specific immune response to some of the drugs that the patients had received during treatment could be established, which was not found in patients with SARS-CoV-2 infection but in those without skin lesions. Interestingly, only two patients had a positive oral challenge: one relative to AZT and one relative to CLA. The discordant results between the oral challenge and LTT could be due to two situations. Firstly, the drugs were not given for a sufficient time or at a sufficient dose during the oral challenge. This may be because the drugs used are so toxic that they cannot be given for long periods without clinical necessity [11]. Alternatively, this may be because the patient was no longer in an inflammatory state that was present during the viral infection. the study was carried out 6 months later; therefore, the patient did not have the cofactor necessary to trigger the cutaneous symptoms again. The 12-month period was chosen for the in vitro study because this is the recommended latency period for LTT studies in severe late drug reactions such as DRESS or exanthema multiforme.

On the other hand, it is important to emphasize the importance of performing the LTT at least 2 months after the resolution of the infection [13,27], with the recommendation being between 6 and 12 months [28]; otherwise, the risk of false-positive results increases, as described previously. Indeed, a case of COVID-19-related cutaneous manifestations has been described in which the proliferation assay was performed 21 days after infection and reported sensitization to all drugs tested [29]. This may be a sign of hyperreactivity caused by a viral infection and, consequently, may be a false-positive LTT result.

Therefore, considering that all patients with late reactions presented skin lesions and had a positive LTT result with a mixed Th1/Th2 cytokine release, it would be reasonable to recommend avoiding the drug in question in all cases. If drug administration is necessary, an exhaustive study under allergological supervision with appropriate dosage and administration time should be carried out. These results highlight the need for a multidisciplinary approach to the management of adverse drug reactions [5].

## 4. Materials and Methods

### 4.1. Lymphocyte Transformation Test (LTT)

The LTT was performed according to Giraldo-Tugores et al. with minor modifications [30]. PBMCs were freshly isolated from heparinized venous blood samples (30 mL) using Ficoll (LymphoPrep™) gradient centrifugation. The cells were resuspended in AIM-V medium (Gibco, Thermo Fisher Scientific Inc., Waltham, MA, USA) (2 × 10^6^ cell/mL) and cultured in 96-well U-bottomed plates (200 µL/well) containing the following stimuli: Dynabeads Human T-Activator CD3/CD28 (1 μL/well) (Gibco) as the positive control; AIM-V or DMSO medium as the negative control (unstimulated condition); and azitromicin (0.1 μg/μL, 1 μg/μL, and 10 μg/μL), amoxicilin (100 μg/μL, 200 μg/μL, and 500 μg/μL), clavulanate (1 μg/μL, 10 μg/μL, and 100 μg/μL), hydroxychloroquine (1 μg/μL, 10 μg/μL, and 100 μg/μL), lopinavir (0.02 μg/μL, 0.1 μg/μL, 0.5 μg/μL, and 2.5 μg/μL), and ritonavir (0.04 μg/μL, 0.2 μg/μL, 1 μg/μL, and 5 μg/μL). Cultures were performed in triplicate and incubated for 4 days in a humidified incubator (37 °C and 5% CO_2_). On day 4, the culture plates were centrifuged, and 100 µL aliquots of the culture supernatant were transferred to another 96-well plate and stored at −40 °C for cytokine analysis. Then, 100 µL of fresh AIM-V medium containing 10 μCi of ^3^H-thymidine (PerkinElmer, Waltham, MA, USA) was added to the cells and gently resuspended on the cell pellet. On day 6, the cultures were transferred to a Multiscreen^®^-HV 96-Well Filter Plate (Merck Millipore, Burlington, MA, USA), and cells were harvested using a MultiScreen^®^Vacuum Manifold (Millipore). Each 96-well filter was punched into a scintillation vial, and radioactivity incorporated into the DNA was measured using a liquid scintillation counter (PerkinElmer). The proliferative response was expressed as a stimulation index (SI), which was calculated using the ratio of disintegrations per minute (dpm) of the drug-stimulated T cells and the mean of dpm of the unstimulated T cells. As part of the standard criteria, an SI > 3 in at least one concentration was considered positive.

### 4.2. Secreted Cytokine Measurement

Four-day cell-culture supernatants were centrifuged and stored at −40 °C. Th1 (IFN-γ, IL-1β, and TNF-α) and Th2 (IL-5, IL-4, IL-6, IL-10, and IL-13) cytokines were measured using the MILLIPLEX^®^ MAP Human High Sensitivity T Cell Magnetic Beads panel (Merck Millipore) according to the manufacturer’s instructions and were acquired on the Luminex Magpix System (Luminex, Austin, TX, USA).

## 5. Conclusions

Using lymphocyte proliferation and cytokine secretion assays, we identified a drug candidate as the culprit of skin lesions during SARS-CoV-2 infection despite only three patients having positive drug provocation test results. Therefore, considering that all patients with late reactions had presented skin lesions and had a positive LTT result with an increase in cytokine secretion, it would be reasonable to recommend avoiding the drug in question in all cases.

## Figures and Tables

**Figure 1 ijms-24-11543-f001:**
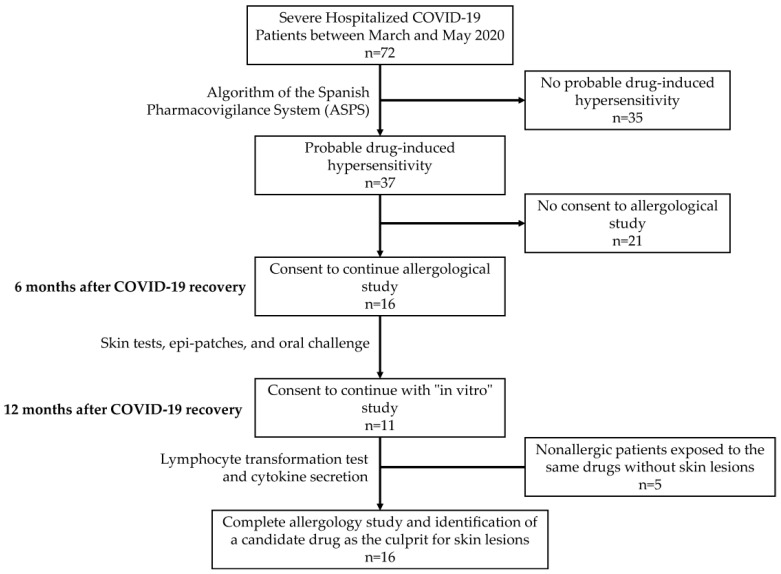
Study design.

**Figure 2 ijms-24-11543-f002:**
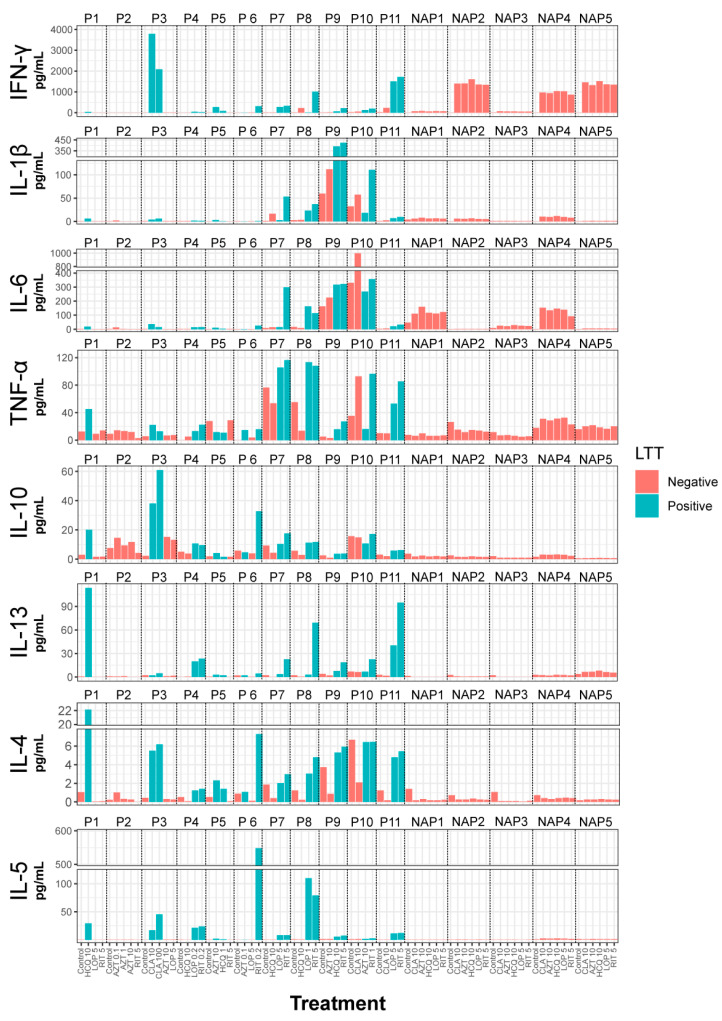
Cytokine secretion of PBMCs in response to all relevant drugs. The red bars correspond to the condition where the LTT result was negative, and the blue bars correspond to the condition where the LTT result was positive.

**Table 1 ijms-24-11543-t001:** Clinical characteristics of the patients and allergological study.

				Epicutaneous Patch Test		Oral Challenges		LTT	
Patient	Age (Years)/ Sex	COVID-19 Treatments	Reaction	Positive	Negative	Positive	Negative	Positive	Negative
P1	60/F	AZT, DOL	MPE ^(1)^		AZT, DOL		AZT, DOL	HCQ	AZT, AMOX, CLA, LOP, RIT
P2	61/M	AZT, KAL, DOL, CEL	UEX ^(1)^		AZT, DOL, KAL, CEL	AZT ^§^	DOL, KAL		HCQ, AZT, AMOX, LOP, RIT
P3	53/M	AZT, KAL, DOL, CEL, AMOX/CLA	MPE ^(1)^		DOL, KAL, CEL, AMOX	AMOX/CLA	KAL, AZT, HCQ	CLA	HCQ, AZT, AMOX, LOP, RIT
P4	63/F	AZT, KAL, DOL, CEL	MPE ^(1)^, VEX ^(2)^		AZT, DOL, KAL		AZT, HCQ, KAL	LOP, RIT	HCQ, AZT, AMOX
P5	66/M	AZT, KAL, DOL, CEL	MPE ^(1)^, VEX ^(2)^		AZT, DOL, KAL, CEL	AZT	KAL, CEL	AZT, HCQ	AMOX, LOP, RIT
P6	61/F	AZT, DOL	MPE ^(1)^ VEX ^(2)^		AZT, DOL		DOL, AZT	AZT, LOP, RIT	HCQ, AMOX
P7	77/F	KAL, DOL, CEL	MPE ^(1)^			*	*	LOP, RIT	HCQ, AZT, AMOX
P8	84/M	AZT, KAL, DOL, CEL	MPE ^(1)^, VEX ^(2)^		AZT, DOL, KAL, CEL		AZT, DOL, KAL, CEL	LOP, RIT	HCQ, AZT, AMOX
P9	76/M	AZT, DOL	VEX ^(2)^		AZT, DOL		AZT, DOL	HCQ, RIT	AZT, AMOX, LOP
P10	74/M	AZT, DOL, CEL	CVAS ^(3)^, CLL ^(4)^		AZT, DOL, CEL	*	*	AZT, RIT	AZT, AMOX, LOP
P11	64/M	AZT, KAL, DOL	MPE ^(1)^		AZT, KAL, DOL		AZT, KAL, DOL	LOP, RIT	HCQ, AZT, AMOX
NAP1	58/M	AZT, KAL, DOL							HCQ, AZT, AMOX, LOP, RIT
NAP2	52/M	AZT, KAL, DOL, CEL							HCQ, AZT, AMOX, LOP, RIT
NAP3	59/M	AZT, KAL, DOL, CEL							HCQ, AZT, AMOX, LOP, RIT
NAP4	49/M	AZT, DOL							HCQ, AZT, AMOX, LOP, RIT
NAP5	70/M	AZT, KAL, DOL, CEL							HCQ, AZT, AMOX, LOP, RIT

^(1)^ Generalized exanthema: MPE—maculopapular exanthem or UEX—urticarial exanthem; ^(2)^ VEX—vesicular exanthem; ^(3)^ CVAS—cutaneous vasculitis; ^(4)^ CLL—chilblain-like lesion. Abbreviations: AZT—azitromicin; AMOX—amoxicillin; CLA—clavulanic acid; DOL—dolquine; HCQ—hydroxychloroquine; KAL—kaletra; LOP—lopinavir; RIT—ritonavir; CEL—ceftriaxona. ^§^ Immediate reaction with 12.5 mg. * Oral challenge not possible because of generalized severe reaction.

**Table 2 ijms-24-11543-t002:** Lymphocyte transformation test (LTT) results.

		Azitromicin			Amoxicillin			Clavulanic Acid			Hidroxiloroquine			Lopinavir			Ritonavir		
Patient	(Dynabeads™ CD3/CD28)	0.1 µg/µL	1 µg/µL	10 µg/µL	100 µg/µL	200 µg/µL	500 µg/µL	1 µg/µL	10 µg/µL	100 µg/µL	1 µg/µL	10 µg/µL	100 µg/µL	0.2 µg/mL	1 µg/mL	5 µg/mL	0.2 µg/µL	1 µg/µL	5 µg/µL
P1	10.2	0.8	1.5	1.5	1.4	1.4	1.2	1.4	1.6	1.6	1.4	**3.2**	0.3	1.7	2.1	2.2	1.4	1.2	1.2
P2	12.2	0.5	1.0	0.5	0.9	1.4	1.5	1.2	1.2	1.4	1.2	1.8	1.9	1.1	0.8	0.9	1.3	1.0	0.8
P3	15.0	1.3	1.5	1.9	1.7	1.7	1.4	1.9	**3.2**	**3.4**	1.4	1.8	0.3	2.0	1.8	0.9	2.0	1.8	1.8
P4	5.8	0.8	0.7	0.9	0.7	2.1	0.7	0.7	0.7	0.8	0.6	1.1	0.1	**3.9**	0.6	0.9	**3.5**	0.3	0.5
P5	9.0	1.7	1.4	**3.1**	1.6	1.4	2.8	1.9	1.7	1.6	**3.4**	0.8	0.7	1.3	1.0	0.8	1.1	1.2	0.6
P6	11.6	**3.0**	1.7	1.0	1.9	1.6	1.9	2.4	2.5	2.3	1.3	1.8	1.6	1.9	**3.0**	2.8	2.9	**3.2**	2.6
P7	9.4	1.4	0.9	1.4	1.8	1.1	1.5	1.8	1.5	1.4	1.7	2.0	1.8	1.9	2.3	**3.4**	2.3	1.0	**3.2**
P8	7.4	1.9	1.7	1.6	1.9	1.8	1.9	1.9	1.5	1.6	1.6	1.9	1.9	1.1	3.3	1.6	1.8	1.5	**3.2**
P9	6.7	1.5	1.3	1.2	1.3	1.2	1.2	1.4	1.3	1.2	1.6	**3.2**	1.5	1.2	2.7	1.8	1.6	**3.0**	**3.4**
P10	8.2	1.1	1.8	**3.1**	0.7	0.9	1.2	1.0	1.3	1.1	0.9	0.8	1.0	2.4	1.5	1.3	2.6	**3.1**	2.1
P11	8.4	1.6	1.4	1.2	1.4	1.2	1.2	1.5	1.8	1.4	1.5	1.3	1.3	1.7	2.1	**3.2**	1.9	2.7	**3.3**
NAP1	6.3	1.4	1.1	1.0	1.1	0.9	0.9	0.9	1.0	0.9	0.4	1.4	0.9	1.1	1.2	1.2	0.9	1.2	0.9
NAP2	5.5	1.9	1.0	1.5	1.3	0.9	0.7	1.6	1.0	0.8	0.4	1.7	0.9	1.8	1.4	1.2	1.8	1.5	1.2
NAP3	5.2	1.5	1.1	1.7	1.9	2.0	1.5	1.9	1.2	1.3	1.8	0.8	1.6	1.0	0.6	0.8	0.9	0.8	0.7
NAP4	9.9	1.8	1.6	2.0	2.0	1.9	1.8	1.9	1.4	1.8	1.9	1.3	1.1	1.5	2.0	1.8	1.9	1.9	1.8
NAP5	7.8	1.4	1.2	0.9	0.8	0.8	0.7	0.9	0.9	0.7	0.3	0.2	1.1	1.9	0.9	0.7	0.9	0.8	0.6

As the standard criteria, an SI ≥ 3 in at least one concentration was considered positive (bold and highlighted in grey).

## Data Availability

Not applicable.

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
