# Peer review of "Value of the Lymphocyte Transformation Test for the Diagnosis of Drug-Induced Hypersensitivity Reactions in Hospitalized Patients with Severe COVID-19"

_ijms, 2023, doi:10.3390/ijms241411543_

Round 1

Reviewer 1 Report

The authors determined the value of the lymphocyte transformation test for the diagnosis of drug-induced hypersensitivity reactions in severe hospitalized COVID-19 patients.

This rare apparently drug related reactions after infection with COVID-19 was well analyzed and described.

Similar late reactions were described previously. The association with COVID-19 appears novel.

The overall importance is not wel described.

In previous publications the term “In the first wave of COVID-19..”

-       Is this phenomenon seen in COVID-19 naïve people?

-       Is this phenomenon only observed in COVID-19 with Wuhan strain or later wirt different strains as well?

-       Is this phenomenon seen with other infections as well?

Author Response

Dear reviewer,

Thank you for your careful review of our paper and for the valuable comments, corrections, and suggestions that have helped us to improve the manuscript. Below, you will find the point-by-point response to your questions.

Reviewer 1

Comments and Suggestions for Authors

The authors determined the value of the lymphocyte transformation test for the diagnosis of drug-induced hypersensitivity reactions in severe hospitalized COVID-19 patients.

This rare apparently drug related reactions after infection with COVID-19 was well analyzed and described.

Similar late reactions were described previously. The association with COVID-19 appears novel.

The overall importance is not well described.

Only a few isolated cases of drug-induced hypersensitivity reactions in patients with COVID-19 have been described in the literature. Some retrospective studies have also been published, but no prospective studies like ours. We recruited 72 patients with skin lesions in the first wave of COVID-19. 37 of them were candidates for drug-induced hypersensitivity, and after a comprehensive allergological study, we were able to find the drug candidate in 11 of them.

To better understand this point, we add a new figure with the study design (Figure 1).

In previous publications the term “In the first wave of COVID-19..”

-       Is this phenomenon seen in COVID-19 naïve people?

The first wave of COVID-19 took place in Spain between March and June 2020. Our patients were recruited between March and May, so they are supposed to be COVID-19 naïve.

-       Is this phenomenon only observed in COVID-19 with Wuhan strain or later wirt different strains as well?

In fact, in Spain, and more specifically in Madrid, in the first wave all patients were infected with the alpha variant, lineages B.1.1.7 and Q. Successive variants have been introduced and coexist. However, after this first wave, the appearance of cutaneous manifestations in patients hospitalised with COVID due to respiratory infection has not been reported with such a high frequency. This finding could be explained by at least two causes: the irruption of new strains that cause a less inflammatory response in the skin, or the type of drugs used in this first wave. As far as the cutaneous inflammatory response is concerned, the alpha variant coexisted with the beta, delta and even gamma variants, so we cannot attribute the reduction in cutaneous manifestations to the viral strain alone. However, there were important changes in the treatment of patients. Our hypothesis is that the combination of virus and drug leads to sensitization of the patient, as has been described in other viral processes.

-       Is this phenomenon seen with other infections as well?

Yes, the association between drug-induced hypersensitivity reactions and viral infections has been described for viruses of the Herpesviridae family, influenza, chikungunya , cytomegalovirus, Epstein-Barr virus or HIV.

To better clarify these two points, we modify the sentence (page 5 line 178) as follows:

“The fact that skin manifestations were greatly reduced in subsequent waves when COVID-19 treatments were changed supports the view that the skin manifestations observed in the first wave were mainly due to hypersensitivity reactions to the drugs used at that time [18] or to the combination of both the drugs used and the viral strain in subsequent waves of COVID-19. Such interactions between drugs and viral infections have been widely reported for viruses of the Herpesviridae family and less commonly for other viruses such as influenza, chikungunya, or HIV [6].”

Reviewer 2 Report

The manuscript represents an interesting approach to analysing skin reactions in patients treated while infected by SARS-CoV-2. The work is a pilot study of 11 patients, and the results encountered are interesting. However, there are some issues to be discussed. The title should be modified, considering it is a pilot study. The second important issue, as referred to in the abstract, is a limitation: some patients, at least 3 of 11, may have had previous reactions to the drugs tested. There is no clear information; perhaps the medical history of previous hypersensibility to AINES or other drugs was incomplete, which should be considered a study limitation. Table 1 is informative, but the age of the patients is a critical issue, and it is lacking in the manuscript. Another point that should be discussed is the lymphocyte response to viral S protein. Do the authors think that should be increased or decreased in these patients? Are they at higher risk of long-COVID?  Looking at the data, and as suggested by several authors, the production of IFN gamma may be decreased in MIS, which would explain why most of the patients did not have a detectable secretion of IFN gamma, IL-2, the critical cytokine is LTT was not analyzed. Patient 3 produces IFN gamma, IL-4 and IL-10 upon the same stimulus, which differs from patient 11, for example. Thus, I would suggest the authors to partially modify the conclusions and include in the discussion a probable explanation of the cytokine responses, but not only based on Th1/Th2 ratio.  

The manuscript was well written. Only minor details were encountered.

Author Response

Dear reviewer,

Thank you for your careful review of our paper and for the valuable comments, corrections, and suggestions that have helped us to improve the manuscript. Below, you will find the point-by-point response to your questions.

Reviewer 2

Comments and Suggestions for Authors

The manuscript represents an interesting approach to analysing skin reactions in patients treated while infected by SARS-CoV-2. The work is a pilot study of 11 patients, and the results encountered are interesting. However, there are some issues to be discussed.

The title should be modified, considering it is a pilot study.

This is not a pilot study, but a continuation of an earlier study in which 72 patients with skin lesions were recruited from the first wave of COVID-1972 patients. Of these, 32 were candidates for drug-induced hypersensitivity. In this work, we have completed the allergological study with LTT and cytokine assays and found a drug candidate in 11 of them.

To better understand this point, we add a new figure with the study design (Figure 1).

The second important issue, as referred to in the abstract, is a limitation: some patients, at least 3 of 11, may have had previous reactions to the drugs tested. There is no clear information; perhaps the medical history of previous hypersensibility to AINES or other drugs was incomplete, which should be considered a study limitation.

None of the patients in this study had a history of drug hypersensitivity prior to COVID-19 infection. In the previous allergology study conducted six months after infection, three patients had positive challenge tests. Of these three patients, one had an immediate hypersensitivity reaction and consequently the LTT was negative. The other two had delayed hypersensitivity reactions confirmed by LTT.

Table 1 is informative, but the age of the patients is a critical issue, and it is lacking in the manuscript.

In addition to Table 1 containing the age of the patients, following your recommendation, we have added the following sentence to the patient description (page 3, line 107):

"Consistent with the general characteristics of the most severe COVID-19 patients, 75% of the patients were male and had a median age of 62 years (IQR 71-58.75)."

Another point that should be discussed is the lymphocyte response to viral S protein.

LTT was performed one year after COVID-19 recovery, so the lymphocytes were not exposed to viral proteins, at the moment of performing the test. It would have been very interesting to co-stimulate the drugs with protein S and see if this would have increased lymphocyte stimulation. However, given the limited amount of sample, it would have been impossible to perform the assay under these conditions, as each patient was tested with six drugs and three concentrations each. This would have meant doubling the amount of sample required.

Do the authors think that should be increased or decreased in these patients?

It would probably be increased lymphocyte proliferation.

Are they at higher risk of long-COVID?

Long-term follow-up of the patients was tracked by medical history and phone call 24 months after the systemic rash, and none of the patients were diagnosed as long-COVID.

Looking at the data, and as suggested by several authors, the production of IFN gamma may be decreased in MIS, which would explain why most of the patients did not have a detectable secretion of IFN gamma, IL-2, the critical cytokine is LTT was not analyzed.

Although it is not as clear in Figure 2 due to the scale, in our study IFN-γ increase with the LTT-positive drugs, as shown more clearly in Supplementary Table 1.

On the other hand, we do not believe that interleukin secretion could be influenced by a possible multisystem inflammatory syndrome, as the LTT and cytokine release assays were performed one year after overcoming the infection to study delayed drug hypersensitivity, precisely to avoid false positives due to the pro-inflammatory state during infection.

Regaring to IL-2, we choose IFN-γ, IL-1β, and TNF-α as a representative of the Th1 response and IL-5, IL-4, IL-6, IL-10, and IL-13 as a representative of the Th2 response. IL-2 appears to play an important role in the early stages of LTT, inducing lymphocyte proliferation at one to two days after T-cell activation.

Patient 3 produces IFN gamma, IL-4 and IL-10 upon the same stimulus, which differs from patient 11, for example.Thus, I would suggest the authors to partially modify the conclusions and include in the discussion a probable explanation of the cytokine responses, but not only based on Th1/Th2 ratio.  

Following your recommendation, we amend the conclusion (page 8, line 290) as follows:

“Therefore, considering all patients with late reactions had presented skin lesions and had a positive LTT result with an increase in cytokine secretion, it would be reasonable to recommend avoiding the drug in question in all cases.”

On the other hand, we do not consider it appropriate to include a probable explanation of the cytokine changes beyond the current discussion (page 5, line 185). In this paragraph, we compare our results with those described by other authors using the same type of LTT assay, which is an ex vivo assay and therefore lacks the immunological context of the organism. LTT assays and the associated cytokine secretion has been shown to be useful in the diagnosis of type IV drug hypersensitivity,  as we show in the present study.

Round 2

Reviewer 1 Report

No further comments

Reviewer 2 Report

The authors made several changes in the manuscript responding to the queries. I still think. However, it is a pilot study since important conclusions can not be made independently of the previous number of patients screened. Due to the clinical interest of the data I consider it important to be published.

Only minor typo errors were encountered